# Human Monkeypox: A Comprehensive Narrative Review and Analysis of the Public Health Implications

**DOI:** 10.3390/microorganisms10081633

**Published:** 2022-08-12

**Authors:** Francesco Di Gennaro, Nicola Veronese, Claudia Marotta, Jae Il Shin, Ai Koyanagi, Andrea Silenzi, Mario Antunes, Annalisa Saracino, Davide Fiore Bavaro, Pinar Soysal, Francesco Vladimiro Segala, Laurie Butler, Eugenio Milano, Mario Barbagallo, Yvonne Barnett, Christopher Parris, Emanuele Nicastri, Damiano Pizzol, Lee Smith

**Affiliations:** 1Clinic of Infectious Diseases, University of Bari, University Hospital Policlinico, 70121 Bari, Italy; 2Geriatric Unit, Department of Internal Medicine and Geriatrics, University of Palermo, 90133 Palermo, Italy; 3General Directorate of Health Prevention, Ministry of Health, 00144 Rome, Italy; 4Department of Pediatrics, Yonsei University College of Medicine, Seoul 03722, Korea; 5Research and Development Unit, Parc Sanitari Sant Joan de Déu, CIBERSAM, ISCIII, 08010 Barcelona, Spain; 6ICREA, Passeig Lluis Companys 23, 08010 Barcelona, Spain; 7Department of Surgery, Catholic University of Mozambique, Beira 13016, Mozambique; 8Department of Geriatric Medicine, Faculty of Medicine, Bezmialem Vakif University, Istanbul 34093, Turkey; 9Faculty of Science and Engineering, Anglia Ruskin University, Cambridge CB1 1PT, UK; 10National Institute for Infectious Diseases, Lazzaro Spallanzani, IRCCS, Via Portuense, 292, 00149 Rome, Italy; 11Italian Agency for Development Cooperation, 00135 Khartoum, Sudan; 12Centre for Health Performance and Wellbeing, Anglia Ruskin University, Cambridge CB1 1PT, UK

**Keywords:** monkeypox (MPX), epidemiology, zoonotic pandemic, smallpox

## Abstract

Recently, numerous cases of monkeypox were reported from several non-endemic countries in Europe, North America, and Oceania, suggesting an unusual and alarming public health issue, particularly considering that the disease is not directly related to human or animal travels. Attention is currently being drawn to this phenomenon since more than 70% of the global population is no longer vaccinated against smallpox. Indeed, the smallpox vaccination also confers some indirect degree of protection against other poxviruses, including monkeypox. We performed a narrative review to describe the existing literature with regard to monkeypox using the MEDLINE, EMBASE, and Scopus databases. This review aims to provide updated evidence of findings on the epidemiology, clinical features, diagnosis, management, and prevention of monkeypox, also considering the concurrent zoonotic pandemic caused by the COVID-19 coronavirus, SARS-CoV-2.

## 1. Introduction

Monkeypox (MPX) is a viral zoonosis brought on by a double-stranded DNA virus. It is a member of the *Poxviridae* family and the orthopoxvirus genus, which also includes the smallpox virus, known as the variola virus. [1]. The virus was initially discovered in monkeys at the Statens Serum Institute in Copenhagen, Denmark, in 1958 (1), giving rise to the sobriquet “monkeypox”; one group of children from the Democratic Republic of the Congo represented the first cases, discovered in 1970 [2]. There are two clades of monkeypox virus (MPXV): the West African variant, with an estimated case fatality ratio (CFR) of less than 4% and with higher prevalence in HIV patients, and the Congo Basin (Central African) variant, with a CFR of 10% [2].

Historically, vaccination against smallpox had been shown to be cross-protective against MPX [3]. However, after the eradication declaration regarding smallpox in 1980 by the World Health Assembly, vaccination against smallpox has ceased and, according to estimates, more than 70% of people worldwide are no longer immune to smallpox.

Since 13 May 2022, many cases of MPX have been reported from several non-endemic countries in Europe, North America, and Australia [4], representing an unusual and alarming public health issue considering that the disease is not directly related to human or animal travels [5]. Indeed, the number of cases reported is surprisingly high; there is no direct link between outbreaks and travels from endemic areas, and it is not clear whether the virus has developed more capacity for transmission between humans, as we know little of its general viral evolution and dynamics [6]. In addition, despite the general low mortality rate and the fact that no death has been reported during the current outbreak, many unusual aspects are creating public concern.

Moreover, although, in recent years, there have been cases reported in non-endemic areas, including the 2003 and 2021 outbreaks in the United States of America and Israel, the September 2018 outbreak in United Kingdom, and the May 2019 outbreak in Singapore, the current global epidemiology pattern has never been recorded before [7].

The MPV genomes are grouped into three monophyletic clades: two previously characterized clades (A.1 and A.2) and a newly emerging clade containing genomes from the ongoing multi-country outbreak in 2022 (B.1) [8,9]. Additionally, a recent article demonstrated the potential function of the enzyme APOBEC3 (host enzyme) in viral growth, as well as in potential MPV adaptability during the course of microevolution, by a detailed investigation of mutation hotspots [10].

Other articles have investigated how nine proteins could be crucial in the pathogenesis of the disease (A9L, A36R, A50L, B9R, B16L, C3L, C7L, C12L (SPI-1), and H5R), while four proteins are crucial for the host’s immune response (A27L, A33R, B5R, and L1R), bringing the scientific community’s attention to their role in disease development and host protection [11].

It is therefore mandatory and urgent to increase research efforts to close the gap of scientific knowledge, in order to stop current and future outbreaks and to optimize our surveillance and preparedness in containing and combatting MPX and zoonotic infections.

This review aims to provide updated evidence of findings on the epidemiology, clinical features, diagnosis, management, and prevention of MPX, also considering the concurrent zoonotic pandemic caused by the coronavirus SARS-CoV-2, which, up until the end of June 2022, has caused almost 600 million COVID-19 cases globally, with 6.3 million deaths [7].

## 2. Methods

We used the following search strategy in Pubmed (“monkeypox”[MeSH Terms] OR “monkeypox”[All Fields] OR (“monkeypox”[MeSH Terms] OR “monkeypox”[All Fields] OR (“monkey”[All Fields] AND “pox”[All Fields]) OR “monkey pox”[All Fields]) OR (“monkeypox virus”[MeSH Terms] OR (“monkeypox”[All Fields] AND “virus”[All Fields]) OR “monkeypox virus”[All Fields] OR “monkeypoxvirus”[All Fields]) OR (“monkeypox virus”[MeSH Terms] OR (“monkeypox”[All Fields] AND “virus”[All Fields]) OR “monkeypox virus”[All Fields]) OR (“monkeypox virus”[MeSH Terms] OR (“monkeypox”[All Fields] AND “virus”[All Fields]) OR “monkeypox virus”[All Fields] OR (“monkey”[All Fields] AND “pox”[All Fields] AND “virus”[All Fields]) OR “monkey pox virus”[All Fields]). The search was then adapted for Scopus for other relevant articles. We included all the observational and intervention studies with human subjects exhibiting MPX, as well as both systematic and nonsystematic reviews.

Since a limited corpus of literature was anticipated, we also incorporated case series, case reports, and pre-prints. The search was complete up to 5 July 2022.

## 3. Epidemiology

In the Democratic Republic of the Congo (DRC), a 9-month-old infant was diagnosed as the first human case in 1970 [12]. Since then, more human MPX cases have been reported, with the DRC seeing the largest increase.

Before an epidemic of 47 confirmed or probable cases that was reported in the US in 2003, which occurred as a result of exposure to infected pet prairie dogs that had contracted the MPX virus from infected exotic animals imported from Ghana, cases of MPX had previously only been recorded in Africa [13,14]. There have been several travel-related cases of MPX in recent years, all following exposure to the disease in Nigeria. In particular, in 2018, there was one case in Israel [15], three in the UK (two in 2018 [16] and one in 2019 [17]), and one in Singapore [18]. A fourth case was reported in the UK in 2018, as a result of nosocomial transmission to a healthcare worker [19].

To date, MPX is considered endemic in the following countries: Benin, Cameroon, the Central African Republic, the Democratic Republic of the Congo, Gabon, Ghana (identified in animals only), the Ivory Coast, Liberia, Nigeria, the Republic of the Congo, Sierra Leone, and South Sudan [20].

Data regarding the incidence of MPX are extremely limited. One study conducted in the Democratic Republic of Congo reported that the incidence increased from 0.64/100,000 in 2001 to 2.82/100,000 in 2013 [21]. Moreover, another study carried out between 2005 and 2007 found that the average annual cumulative incidence of confirmed MPX cases from nine health zones in the Sankuru district was 0.55 per 100,000 [22].

Data on case fatality rates were retrieved from a recent literature review that calculated a pooled estimate case fatality ratio (CFR) of 8.7% across all countries [23].

However, it is of note that all the deaths were observed in Africa [24]. In the same study, when the data were analyzed according to clade, the CFR for the Central African clade (10.6%, 95% CI: 8.4–13.3%) was significantly higher than that for the West African clade (3.7%, 95% CI: 1.7–6.8%) [25].

Considering the demographic characteristics, males are more affected than females; this condition primarily affects young people, despite the fact that the median age at presentation increased from 4–5 years in 1970–1989 to 10 years in 2000–2009 and 21 years in 2010–2019 [22,23,24].

## 4. Monkeypox Multi-Country Outbreak, 2022

Between 1 January and 15 June 2022, 42 nations in five WHO regions collectively reported 2103 laboratory-confirmed cases, with only one death [25]. Most cases (98%) that have been documented since May 2022 have been identified in men who have sex with men (MSM), who are seeking care in primary care and sexual health clinics; however, this is not always the case. To yet, no travel connections to endemic regions have been discovered.

Up until 5 July 2022, 5949 cases of MPX had been identified across the European region via IHR mechanisms and official public resources from 33 countries. Of the 5266 cases reported in the European surveillance system (TESSy), 5265 had laboratory confirmation, and 99 were confirmed to be of the West African clade where sequencing was available. The earliest reported date of symptom onset was 17 April 2022. The majority of cases were in individuals who were between the ages of 31 and 40 (2214/5258-42%) and were male (5209/5230-99.6%). Of those cases with known HIV status, 40% (364/917) were HIV-positive [26]. The majority of patients (2684/2793; 96.1%) presented with a rash, while 1931/2793; 69% had systemic symptoms such as fever, exhaustion, muscular discomfort, vomiting, diarrhea, chills, sore throat, or headache [23]. There were no recorded deaths in any of the cases [27]. Although some (15) instances involving health professionals were recorded, more research is being conducted to ascertain whether the infections were caused by exposure at work [26,27]. Given the absence of epidemiological links to endemic areas, the unexpected appearance of MPX in several regions suggests that undetected transmission may have been occurring for a considerable time.

The current outbreak varies from the previous ones in terms of age (most of the people affected are in their thirties), sex/gender (most cases are male), risk factors, and mode of transmission, with sexual transmission being very likely. Along with being characterized by anogenital lesions and rashes that mostly spare the face and limbs, the clinical appearance is also uncommon and distinctive. Fever, lymphadenopathy, exanthema, asthenia, weariness, and headache were the most common signs and symptoms [28,29].

### 4.1. Pathophysiology and Clinical Manifestation

The MPX’s natural reservoir has yet to be established, while rats are the most likely suspect. A possible risk factor is eating undercooked meat and other animal products from infected animals. People who live in or near wooded regions may also be exposed to infected animals in an indirect or low-level manner. Although MPX is not easily transmitted, it can be transmitted through contact with contaminated body fluids or lesion materials, both directly and indirectly [26]. Direct exposure includes contact with fomites, respiratory secretions, or skin-to-skin contact with MPX patients. Being in the patient’s room or within 6 feet of a patient while they are undergoing any treatments that might produce aerosols from oral secretions, skin lesions, or the resuspension of dried exudates without donning an N95 mask and eye protection can result in indirect exposure [27]. Transmission can also happen through the placenta (which can cause congenital MPX) or through intimate contact during and after child delivery.

After entry, the virus replicates at the inoculation site, first localizing in mononuclear phagocytic cells. Then, it is released into the bloodstream and, finally, localizes again in skin cells. Following the first step of replication, it spreads to local lymph nodes and, thus, provokes a viremia within 10 to 14 days (the possible incubation period) [28].

The characteristic clinical manifestation, consisting of a vesiculo-pustular rash, is usually preceded by prodromal non-specific symptoms, such as fever, chills, myalgia, headache, lethargy, and lymphadenopathy [29]. Importantly, patients are infectious starting from the prodromal symptoms until the lesions form scabs and the scabs fall off. Usually, the oropharynx is the first site affected, after which lesions appear on the skin.

The clinical presentation of MPX cases associated with the current outbreak has been variable thus far. Many cases in this outbreak do not exhibit the classically described clinical patterns for MPX (fever and swollen lymph nodes, followed by a centrifugal evolving rash). The presence of only a few or even a single lesion, lesions that start in the genital or perineal/perianal region and do not spread, lesions that manifest at various (asynchronous) stages of development, and the appearance of lesions before the onset of fever, malaise, or other constitutional symptoms are some of the examples of abnormal characteristics [29,30]. The mechanisms of transmission during sexual contact remain unknown, despite the fact that it is known that close physical and personal skin-to-skin or face-to-face contact might result in transmission (through direct contact with infected skin).

### 4.2. Diagnosis and Differential Diagnosis

For the diagnosis of MPX, it is essential to first define the ‘suspect case’ in accordance with WHO recommendations, i.e., a person of any age who presents in a non-MPX endemic country with an unexplained acute rash, and with one or more of the following signs or symptoms for which the common causes of acute eruption do not explain the clinical presentation: headache, the acute onset of a fever above 38.5 °C, lymphadenopathy, myalgia, back pain, and asthenia [16]. In addition, many authors report a characteristic triad for the diagnosis of MPX: skin lesions, lymphadenomegaly, and fever [30,31,32].

For its specificity and sensitivity, the polymerase chain reaction (PCR) test is the gold standard laboratory test, but the type and quality of the sample for the laboratory test is crucial. As a result, the best diagnostic samples for MPX are fluid from vesicles and pustules, as well as dried scabs. Moreover, when it is possible, a biopsy may be utilized. As per the CDC recommendations, the lesion samples must be maintained in a cool environment and stored in a dry, sterile tube [20]. Experience has shown that MPX virus DNA may be found in saliva, blood, urine, semen, feces, and nasopharyngeal swabs [30,31,32].

Unfortunately, antigen and antibody detection assays do not offer MPX-specific confirmation because orthopoxviruses are serologically cross-reactive. When resources are limited, serology and antigen detection procedures are not recommended for diagnosis or case investigation. Furthermore, recent or past vaccine-based immunization may result in misleading positive findings. In addition, PCR tests on pharyngeal swabs and seminal fluid may be a good strategy, considering the transmission pathways of the virus, especially if the patient presents symptoms such as a sore throat or penile lesions.

As part of a clinical differential diagnosis, it is important to rule out other rash-presenting conditions, such as molluscum contagiosum, chickenpox, measles, bacterial skin infections, scabies, syphilis, and medication-related allergies.

Lymphadenopathy can be used to differentiate MPX from chickenpox or smallpox during the prodromal stage of sickness. Additionally, non-infectious illnesses, including Behcet’s diseases, squamous cell carcinoma, and recurrent aphthous stomatitis must be ruled out, in addition to other MST conditions (sexually transmitted diseases), such as the herpes simplex virus, syphilis, chancroid, lymphogranuloma venereum (LGV), and granuloma inguinale are crucial in differential diagnosis [32,33].

### 4.3. Prevention and Treatment

Vaccination with first-generation (e.g., Dryvax, Aventis Pasteur Smallpox Vaccine), second-generation (e.g., ACAM2000) and third-generation (IMVAMUNE and LC16m8) vaccines is the first line of defense against orthopoxvirus disease. Despite the fact that these vaccines provide effective protection, their widespread use is limited by the high rate of adverse events associated with live, attenuated virus immunization [34,35].

All first- and second-generation vaccines use a live, replication-competent virus. A successful vaccination procedure produces a lesion at the site of administration that generates the infectious virus. However, there is the risk of adverse events, such as autoinoculation to other parts of the body, as well as inadvertent transmission to other people, post-vaccine encephalitis, and disseminated infection in immunosuppressed patients. The third-generation vaccine no longer poses the risk of autoinoculation, unintentional transmission, or systemic dissemination, while having a better safety profile due to its decreased capacity to replicate in mammalian cells and the absence of a lesion at the site of immunization. It is recommended for people who have higher risk factors for adverse outcomes, such as people with HIV, patients with hematological conditions, and immunocompromised individuals [35,36,37].

There are no recognized treatment guidelines for MPX infection at this time.

Tecovirimat, a viral envelope protein p37 inhibitor that prevents virus particles from being released from infected cells, has been shown to be beneficial in treating a range of poxvirus-related illnesses in animal studies. It has been approved by the US Food and Drug Administration (FDA) (Silver Spring, MD, USA) for the treatment of smallpox since July 2018, and by the European Medicines Agency (EMA) (Amsterdam, The Netherlands) for the treatment of MPX and cowpox since January 2022, despite the lack of clinical efficacy trials [38,39]. A very recent study on seven people with MPX in the UK in the period from 2018 to 2021 showed good profile efficacy with Tecovirimat (ST-246) [40]. Currently, Tecovirimat may be considered for treatment in people suffering from severe disease (e.g., hemorrhagic manifestation, CNS involvement, confluent lesions, and sepsis) or in people at a high risk of severe disease (people with immunocompromised conditions or pediatric populations, particularly in patients younger than 8 years of age or in pregnant or breastfeeding women).

Although there is inadequate data on the efficiency of treating MPX in humans, the antivirals cidofovir (CDV) and brincidofovir (BCV) might be utilized to treat MPX. These inhibit viral DNA polymerase and have variable degrees of effectiveness in treating various viral infections.

CDV and BCV have previously shown antiviral efficacy against adenoviruses and poxviruses, among other double-stranded DNA viruses. BCV is a lipid compound of the nucleotide analog cidofovir, also known as hexadecyloxypropyl-cidofovir [HDP-CDV] or CMX001 (CDV). BCV has a higher cellular absorption rate and demonstrates greater conversion to the active form by intracellular enzymes than CDV.

Some studies have shown an interesting profile of BCV in animal models, reducing mortality and the risk of disease progression. In a mouse model, Brincidofovir (CMX001) and the smallpox vaccination can be given together without compromising immunity [41].

### 4.4. Public Health Response

Although there are different hypotheses on how MPX reached non-endemic countries during the current outbreak [42], it appears to be spreading disproportionately among men who have sex with men (MSM), meeting via social networks [43,44]. However, the complexities of the current MPX outbreak in the context of non-endemic countries necessitate a comprehensive response that combines traditional public health countermeasures with risk communication and community strategies to engage diverse audiences, based on the different risks of exposure but without stigmatization [45,46].

From a public health perspective, the priority should be to contain the virus’s spread. This could be achieved by implementing specific actions in both healthcare and community settings. In healthcare settings, increased vigilance and the clinical recognition of disease symptoms are essential to ensure the early detection of cases, along with the notification and isolation of patients. Healthcare human resources departments should be trained and equipped with all the diagnostic tools needed to make a timely diagnosis and isolate the suspected or confirmed cases in places with adequate ventilation; a dedicated bathroom is strongly recommended. In the presence of a clinical picture that does not require hospitalization-if housing and hygiene conditions permit, the confirmed case may be monitored at home according to locally defined procedures, also in isolation from cohabitants and any other caregivers.

In addition, protecting healthcare workers and preventing transmission in healthcare settings (using PPE and following infection prevention and control procedures) should be guaranteed as soon as the outbreak is recognized. Healthcare workers caring for patients with suspected or confirmed MPX should implement standard contact and droplet precautions, both in outpatient and hospital settings. Hand hygiene must be strictly followed, contaminated medical equipment must be handled with care, laundry and garbage must be disposed of properly, and environmental surfaces must be cleaned and disinfected.

In the community, in order to stop the infection’s chains of transmission, it would be important to intensify surveillance in specific population groups, such as MSM communities during the current outbreak, along with case and cluster investigation and contact-tracing activities. Indeed, surveillance and case investigation for MPX in the current context are essential to rapidly identify cases, clusters, and sources of infection as early as possible, in order to provide optimal clinical care, isolate cases to prevent further transmission, identify and manage contacts and adapt efficient control and preventive strategies based on the most frequently found transmission channels. Due to the risk to the public health posed by even one MPX case, which is considered an outbreak in itself in non-endemic countries, suspected cases should be reported to local and national health authorities immediately, regardless of whether other potential diagnoses are also being investigated. Cases should be reported as soon as possible, using the case definitions shared by public health authorities.

The surveillance of close contacts is also recommended, i.e., self-monitoring daily for the occurrence of signs/symptoms associated with MPX for a period of 21 days after the last contact with a patient or their contaminated materials during the infectious period. During the surveillance period, MPX infection contacts should avoid encounters with immunocompromised persons, pregnant women, and children under 12 years of age, and should not donate blood, cells, tissues, organs, breast milk, or semen.

As for vaccines and immunization, although smallpox infection vaccines are thought to provide protection against MPX, using them during the 2022 outbreak is unlikely. Because of the cessation of smallpox vaccination programs over the last 50 years, as well as a lack of supply of effective vaccines, which are mostly only available in a few national stockpiles, these countermeasures, which are particularly suitable in the case of contact prophylaxis, are not readily available in the short term.

Thus, implementing effective communication strategies should be a priority. Risk communication and community engagement are key measures. In fact, the primary preventative approach for MPX is to increase the knowledge of risk factors and inform individuals of the steps they may take to lessen exposure to the virus. Community engagement can also help to avoid the stigmatization of at-risk population groups and reduce the impact on societies and economics [47]. Public health authorities should always apply measures that are commensurate to the risk. At the current stage, international travel-related measures are not recommended but it could be important to promote safe mass-gathering events in areas experiencing outbreaks [48]. In addition, public health containment measures, which are generally accepted in MPX-endemic countries, may not be as widely accepted in Western countries, especially after the COVID-19 pandemic. For example, a recent case of MPX school transmission, observed in Quebec, may generate debate and difficulties for public health professionals that should be prevented by timely and specific institutional communication [49].

Learning from the COVID-19 pandemic, from a global perspective, governments, agencies, NGOs, and industries should continue to share information, diagnostic resources, and data, as well as collaborate to ensure equitable access to countermeasures (vaccine, therapeutics, and diagnostics) based on public health needs.

Strategies would also be to strengthen a “one health” approach in endemic countries, as environmental factors increase the frequency of contact with potential hosts, raising the risk of animal-to-human transmission.

## 5. Conclusions

This review gives readers insight into the current MPX scenario and paints a picture of the state of the art in terms of the influence on and reaction to public health, epidemiology, clinical features, diagnosis, management, treatment, and prevention. Although MPX is not a new virus and has appeared in regular outbreaks in poorer and remote areas in Central and West Africa, the literature on this topic is limited. MPX is a perfect example of the potentially explosive mix of zoonotic spillover and anthropogenic drivers that accounts for the vast majority of the world’s epidemic potential. Unlike other neglected diseases, we have options for treating and preventing MPX infection, but access is challenging in that part of the world where it is most needed. Preparing for epidemics and global health challenges requires responding to ongoing outbreaks where and when they occur, rather than waiting for them to spread elsewhere. The growing scientific attention may contribute to a better understanding of this infection and to the identification of the gold standard in terms of prevention, treatment, and management; however, it will be important that each step in scientific progress and medical innovation can also be accessible to those who can benefit the most. This is critical, not only for equity but also for global health security, because it is in everyone’s best interest to solve such a problem before it becomes a more major one.

As for the current multi-country outbreak, some challenges and open questions, such as the unusually high number of cases in several countries in a short timeframe, the uncertainty about the future evolution of the disease, and the current limited existing medical countermeasures, still remain. Furthermore, since smallpox infection vaccines provide protection against MPX, and the vaccine is ready, we encourage the utilization of vaccines as soon as possible, at least for specific population groups and healthcare workers.

Finally, this outbreak is bringing back the specter of syndemic risk that has been summoned by the recent and ongoing COVID pandemic. Thus, it is mandatory to increase research efforts in order to block the infectious risk, not only in the short term but also in the long term and for all countries of the world.

## Data Availability

Not applicable.

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
