# Peer review of "Human Monkeypox: A Comprehensive Narrative Review and Analysis of the Public Health Implications"

_microorganisms, 2022, doi:10.3390/microorganisms10081633_

Round 1

Reviewer 1 Report

This is an interesting review describing information of findings on epidemiology, clinical features, diagnosis, management, and prevention of monkeypox also considering the concurrent zoonotic pandemic caused by the coronavirus SARS-CoV-2. Unfortunately, the manuscript needs further editing and corrections to make it suitable for journal publication. One significant aspect of the paper which needs substantial attention relates to references used. I will provide some examples that can enrich the manuscript. 

1.Introduction

            Include a paragraph on the genetic findings of the current outbreak lineages. 

Luna N, Ramírez AL, Muñoz M, Ballesteros N, Patiño LH, Castañeda SA, Bonilla-Aldana DK, Paniz-Mondolfi A, Ramírez JD. Phylogenomic analysis of the monkeypox virus (MPXV) 2022 outbreak: Emergence of a novel viral lineage? Travel Med Infect Dis. 2022 Jul 13;49:102402. doi: 10.1016/j.tmaid.2022.102402. Epub ahead of print. PMID: 35840078.

Mayla Abrahim, Alexandro Guterres, Patrícia Cristina da Costa Neves, Ana Paula Dinis Ano Bom. The emergence of new lineages of the Monkeypox virus could affect the 2022 outbreak. bioRxiv 2022.07.07.498743;  doi: https://doi.org/10.1101/2022.07.07.498743.

Isidro J, Borges V, Pinto M, Sobral D, Santos JD, Nunes A, Mixão V, Ferreira R, Santos D, Duarte S, Vieira L, Borrego MJ, Núncio S, de Carvalho IL, Pelerito A, Cordeiro R, Gomes JP. Phylogenomic characterization and signs of microevolution in the 2022 multi-country outbreak of monkeypox virus. Nat Med. 2022 Jun 24. doi: 10.1038/s41591-022-01907-y. Epub ahead of print. PMID: 35750157.

Babkin IV, Babkina IN, Tikunova NV. An Update of Orthopoxvirus Molecular Evolution. Viruses. 2022 Feb 14;14(2):388. doi: 10.3390/v14020388. PMID: 35215981; PMCID: PMC8875945.

Methods

I would encourage authors to consider the inclusion of preprints, with good judgment. 

4. Monkeypox multi-country outbreak, 2022.

            Pathophysiology ans clinical manifestation.

           I would encourage authors to consider the inclusion and discuss of all published articles.

Bragazzi NL, Kong JD, Mahroum N, Tsigalou C, Khamisy-Farah R, Converti M, Wu J. Epidemiological trends and clinical features of the ongoing monkeypox epidemic: A preliminary pooled data analysis and literature review. J Med Virol. 2022 Jun 12. doi: 10.1002/jmv.27931. Epub ahead of print. PMID: 35692117.

Thornhill JP, Barkati S, Walmsley S, Rockstroh J, Antinori A, Harrison LB, Palich R, Nori A, Reeves I, Habibi MS, Apea V, Boesecke C, Vandekerckhove L, Yakubovsky M, Sendagorta E, Blanco JL, Florence E, Moschese D, Maltez FM, Goorhuis A, Pourcher V, Migaud P, Noe S, Pintado C, Maggi F, Hansen AE, Hoffmann C, Lezama JI, Mussini C, Cattelan A, Makofane K, Tan D, Nozza S, Nemeth J, Klein MB, Orkin CM; SHARE-net Clinical Group. Monkeypox Virus Infection in Humans across 16 Countries - April-June 2022. N Engl J Med. 2022 Jul 21. doi: 10.1056/NEJMoa2207323. Epub ahead of print. PMID: 35866746.

Diagnosis and differential diagnosis

Peiró-Mestres A, Fuertes I, Camprubí-Ferrer D, Marcos MÁ, Vilella A, Navarro M, Rodriguez-Elena L, Riera J, Català A, Martínez MJ, Blanco JL; Hospital Clinic de Barcelona Monkeypox Study Group. Frequent detection of monkeypox virus DNA in saliva, semen, and other clinical samples from 12 patients, Barcelona, Spain, May to June 2022. Euro Surveill. 2022 Jul;27(28). doi: 10.2807/1560-7917.ES.2022.27.28.2200503. PMID: 35837964.

Aden TA, Blevins P, York SW, Rager S, Balachandran D, Hutson CL, Lowe D, Mangal CN, Wolford T, Matheny A, Davidson W, Wilkins K, Cook R, Roulo RM, White MK, Berman L, Murray J, Laurance J, Francis D, Green NM, Berumen RA 3rd, Gonzalez A, Evans S, Hudziec M, Noel D, Adjei M, Hovan G, Lee P, Tate L, Gose RB, Voermans R, Crew J, Adam PR, Haydel D, Lukula S, Matluk N, Shah S, Featherston J, Ware D, Pettit D, McCutchen E, Acheampong E, Buttery E, Gorzalski A, Perry M, Fowler R, Lee RB, Nickla R, Huard R, Moore A, Jones K, Johnson R, Swaney E, Jaramillo J, Reinoso Webb C, Guin B, Yost J, Atkinson A, Griffin-Thomas L, Chenette J, Gant J, Sterkel A, Ghuman HK, Lute J, Smole SC, Arora V, Demontigny CK, Bielby M, Geeter E, Newman KAM, Glazier M, Lutkemeier W, Nelson M, Martinez R, Chaitram J, Honein MA, Villanueva JM. Rapid Diagnostic Testing for Response to the Monkeypox Outbreak - Laboratory Response Network, United States, May 17-June 30, 2022. MMWR Morb Mortal Wkly Rep. 2022 Jul 15;71(28):904-907. doi: 10.15585/mmwr.mm7128e1. PMID: 35834423.

An accurate and rapid Real-time PCR approach for human Monkeypox virus diagnosis Tony Wawina-Bokalanga, Nikola Sklenovska, Bert Vanmechelen, Mandy Bloemen, Valentijn Vergote, Lies Laenen, Emmanuel André, Marc Van Ranst, Jean-Jacques Muyembe-Tamfum, Piet Maes medRxiv 2022.06.23.22276033; doi: https://doi.org/10.1101/2022.06.23.22276033

Author Response

To Microorganisms editor,

We have appreciated the positive feedback to our manuscript “Human Monkeypox: A Comprehensive and Public Health implications”.

We have considered all the useful suggestions made by the referees and we have implemented the text. We have also satisfied the technical requirements according to the journal guidelines. Modifications have been highlighted using the "track changes" feature. Also, a native English speaker has been engaged to improve the fluency and the readability of the manuscript.

We believe that the revision proposed by the referees, and further implemented in the text, contributed to improve the manuscript. Thus, we kindly ask you to re-consider the manuscript for publication.

Please find a point-by-point response to the referees’ comments below.

Best regards,

Prof. Francesco Di Gennaro

Reviewer 1

This is an interesting review describing information of findings on epidemiology, clinical features, diagnosis, management, and prevention of monkeypox also considering the concurrent zoonotic pandemic caused by the coronavirus SARS-CoV-2. Unfortunately, the manuscript needs further editing and corrections to make it suitable for journal publication. One significant aspect of the paper which needs substantial attention relates to references used. I will provide some examples that can enrich the manuscript. 

Response: We thank you very much for the encouraging feedback on our manuscript. A native English speaker has been engaged to improve the fluency and the readability of the manuscript. We included all references that you suggested

1.Introduction

  Include a paragraph on the genetic findings of the current outbreak lineages. 

Luna N, Ramírez AL, Muñoz M, Ballesteros N, Patiño LH, Castañeda SA, Bonilla-Aldana DK, Paniz-Mondolfi A, Ramírez JD. Phylogenomic analysis of the monkeypox virus (MPXV) 2022 outbreak: Emergence of a novel viral lineage? Travel Med Infect Dis. 2022 Jul 13;49:102402. doi: 10.1016/j.tmaid.2022.102402. Epub ahead of print. PMID: 35840078.

Mayla Abrahim, Alexandro Guterres, Patrícia Cristina da Costa Neves, Ana Paula Dinis Ano Bom. The emergence of new lineages of the Monkeypox virus could affect the 2022 outbreak. bioRxiv 2022.07.07.498743;  doi: https://doi.org/10.1101/2022.07.07.498743.

Isidro J, Borges V, Pinto M, Sobral D, Santos JD, Nunes A, Mixão V, Ferreira R, Santos D, Duarte S, Vieira L, Borrego MJ, Núncio S, de Carvalho IL, Pelerito A, Cordeiro R, Gomes JP. Phylogenomic characterization and signs of microevolution in the 2022 multi-country outbreak of monkeypox virus. Nat Med. 2022 Jun 24. doi: 10.1038/s41591-022-01907-y. Epub ahead of print. PMID: 35750157.

Babkin IV, Babkina IN, Tikunova NV. An Update of Orthopoxvirus Molecular Evolution. Viruses. 2022 Feb 14;14(2):388. doi: 10.3390/v14020388. PMID: 35215981; PMCID: PMC8875945.

Response: Many thanks for your important suggestions. We read the paper with great interest and add this prepreint in our paper. Thanks for suggestions

Methods

I would encourage authors to consider the inclusion of preprints, with good judgment. 

Response: Thanks. We include also preprint in our search strategy

  1. Monkeypox multi-country outbreak, 2022.  Pathophysiology ans clinical manifestation.

           I would encourage authors to consider the inclusion and discuss of all published articles.

Bragazzi NL, Kong JD, Mahroum N, Tsigalou C, Khamisy-Farah R, Converti M, Wu J. Epidemiological trends and clinical features of the ongoing monkeypox epidemic: A preliminary pooled data analysis and literature review. J Med Virol. 2022 Jun 12. doi: 10.1002/jmv.27931. Epub ahead of print. PMID: 35692117.

Thornhill JP, Barkati S, Walmsley S, Rockstroh J, Antinori A, Harrison LB, Palich R, Nori A, Reeves I, Habibi MS, Apea V, Boesecke C, Vandekerckhove L, Yakubovsky M, Sendagorta E, Blanco JL, Florence E, Moschese D, Maltez FM, Goorhuis A, Pourcher V, Migaud P, Noe S, Pintado C, Maggi F, Hansen AE, Hoffmann C, Lezama JI, Mussini C, Cattelan A, Makofane K, Tan D, Nozza S, Nemeth J, Klein MB, Orkin CM; SHARE-net Clinical Group. Monkeypox Virus Infection in Humans across 16 Countries - April-June 2022. N Engl J Med. 2022 Jul 21. doi: 10.1056/NEJMoa2207323. Epub ahead of print. PMID: 35866746.

Diagnosis and differential diagnosis

Peiró-Mestres A, Fuertes I, Camprubí-Ferrer D, Marcos MÁ, Vilella A, Navarro M, Rodriguez-Elena L, Riera J, Català A, Martínez MJ, Blanco JL; Hospital Clinic de Barcelona Monkeypox Study Group. Frequent detection of monkeypox virus DNA in saliva, semen, and other clinical samples from 12 patients, Barcelona, Spain, May to June 2022. Euro Surveill. 2022 Jul;27(28). doi: 10.2807/1560-7917.ES.2022.27.28.2200503. PMID: 35837964.

Aden TA, Blevins P, York SW, Rager S, Balachandran D, Hutson CL, Lowe D, Mangal CN, Wolford T, Matheny A, Davidson W, Wilkins K, Cook R, Roulo RM, White MK, Berman L, Murray J, Laurance J, Francis D, Green NM, Berumen RA 3rd, Gonzalez A, Evans S, Hudziec M, Noel D, Adjei M, Hovan G, Lee P, Tate L, Gose RB, Voermans R, Crew J, Adam PR, Haydel D, Lukula S, Matluk N, Shah S, Featherston J, Ware D, Pettit D, McCutchen E, Acheampong E, Buttery E, Gorzalski A, Perry M, Fowler R, Lee RB, Nickla R, Huard R, Moore A, Jones K, Johnson R, Swaney E, Jaramillo J, Reinoso Webb C, Guin B, Yost J, Atkinson A, Griffin-Thomas L, Chenette J, Gant J, Sterkel A, Ghuman HK, Lute J, Smole SC, Arora V, Demontigny CK, Bielby M, Geeter E, Newman KAM, Glazier M, Lutkemeier W, Nelson M, Martinez R, Chaitram J, Honein MA, Villanueva JM. Rapid Diagnostic Testing for Response to the Monkeypox Outbreak - Laboratory Response Network, United States, May 17-June 30, 2022. MMWR Morb Mortal Wkly Rep. 2022 Jul 15;71(28):904-907. doi: 10.15585/mmwr.mm7128e1. PMID: 35834423.

An accurate and rapid Real-time PCR approach for human Monkeypox virus diagnosis Tony Wawina-Bokalanga, Nikola Sklenovska, Bert Vanmechelen, Mandy Bloemen, Valentijn Vergote, Lies Laenen, Emmanuel André, Marc Van Ranst, Jean-Jacques Muyembe-Tamfum, Piet Maes medRxiv 2022.06.23.22276033; doi: https://doi.org/10.1101/2022.06.23.22276033

Response: dear Reviewer, many thanks for your relevant suggestions. We appreciate the paper that you suggested and we included them in our references

Reviewer 2 Report

This article Human Monkeypox: A Comprehensive and Public Health implications reviewed related study and data of MPX from Pubmed, discussed from its history to the recent outbreak; from its pathophysiology to the diagnosis; from the prevention and treatment to the public health response. The demonstration of the article is comprehensive with humanistic care.

There are some questions need to be answered and changed as below.

1.       “However, of note all the deaths were observed in Africa [20]. In the same study, when 107

the data were analyzed by clade, the CFR for the Central African clade (10.6%, 95% CI: 108

8.4–13.3%) was significantly higher than that of the West African clade (3.6%, 95% CI: 1.7– 109

6.8%) [19].”

“There are two clades of monkeypox virus (MPXV): the 40

West African, with an estimated case fatality ratio (CFR) less than 1% and higher 41

prevalence in HIV patients, and the Congo Basin (Central African) with a CFR of 10% [2]. 42”

The case fatality ratio number of West African in the introduction (Line 109) isn’t consistent with the number (Line 41) in Epidemiology section. Please indicate the year of data and why it is different.

2.       “Considering the demographic characteristics, males are more affected than females, 111

and this condition primarily affects young people, despite the fact that the median age at 112

presentation increased from 4-5 years in 1970-1989 to 10 years in 2000-2009 and 21 years 113

in 2010-2019. [18-20]”

The trend of the presentation of median age is obvious (Line112-113). But it is unclear what’s the median age of each group. It will be more informative to demonstrate the age of young people who got infected in different period of years in detail.

3.       “Unfortunately, antigen and antibody detection assays do not offer MPX -specific 182

confirmation because orthopoxviruses are serologically cross-reactive. When resources 183

are limited, serology and antigen detection procedures are not recommended for 184

diagnosis or case investigation. Furthermore, recent or past vaccine-based immunization 185

may result in misleading positive findings. In addition, PCR on pharyngeal swabs and 186

seminal fluid may be a good strategy considering the transmission pathways of the virus, 187

especially if the patient presents symptoms such as sore throat or penile lesions.”

It is necessary to reference articles to prove antigen and antibody detection assays do not offer MPX -specific confirmation, and recent or past vaccine-based immunization may result in misleading positive findings ( Line185,186)

4.       “Vaccination with first-generation (e.g. Dryvax, Aventis Pasteur Smallpox Vaccine), 225

second-generation (e.g. ACAM2000) and third-generation (IMVAMUNE and LC16m8) 226

vaccines is the first line of defense against orthopoxvirus disease. Despite the fact that 227

these vaccines provide great protection, their widespread use is limited by the high rate 228

of adverse events associated with live, attenuated virus immunization. 229

All first- and second-generation vaccines use live, replication-competent virus. A 230

successful vaccination produces a lesion at the site of administration that generates 231

infectious virus. However, there is risk of adverse events such as autoinoculation to other 232

parts of the body as well as inadvertent transmission to other people, postvaccine 233

encephalitis, and disseminated infection in immunodepressed patients. While third 234

generation vaccine has a higher safety profile owing to the lost ability to replicate in 235

mammalian cells and non-production of a lesion at the site of vaccination, it no longer 236

presents a risk of autoinoculation, inadvertent transmission, or systemic spread. It is 237

reserved for persons with increased risk factors for adverse events such as HIV, 238

haematological and immunosuppressed patients. [34-35]”

I would suggest to discuss about prevention (Line225-238) part before treatment (Line199-224) in “Prevention and Treatment” section

5.       “In the community, in order to stop the chains of transmission, it would be important 266

to intensify surveillance in specific population groups – such as MSM communities during 267

the current outbreak -, along with case and cluster investigation and contact tracing 268

activities. Indeed, surveillance and case investigation for MPXV in the current context are 269

essential to rapidly identify cases, clusters and sources of infection as early as possible in 270

order to provide optimal clinical care, isolate cases to prevent further transmission, 271

identify and manage contacts, adapt effective prevention methods and control based on 272

the most commonly identified routes of transmission.

Please correct some wording (“MPXV” to “MPX”) and symbol (“current outbreak “- “ to “–“  ) mistakes.

6.       As for vaccines and immunization, although smallpox infection vaccines are thought 285

to provide protection against MPX, using them during the 2022 outbreak is unlikely. 286

Because of the cessation of smallpox vaccination programs over the last 50 years, as well 287

as a lack of supply of effective vaccines, which are mostly only available in a few national 288

stockpiles, these countermeasures, particularly suitable for contact prophylaxis, are not 289

readily available in the short term.

 I understand it isn’t readily available in the short term. Since smallpox infection vaccines provided protection against MPX, and the technique is ready, it should be encouraged to product those vaccines as soon as possible, at least for specific population groups.

7.       The logic of demonstration in each paragraph needs to be improved. There are many paragraphs that saying the same topic could be combined.

Author Response

To Microorganisms editor,

We have appreciated the positive feedback to our manuscript “Human Monkeypox: A Comprehensive and Public Health implications”.

We have considered all the useful suggestions made by the referees and we have implemented the text. We have also satisfied the technical requirements according to the journal guidelines. Modifications have been highlighted using the "track changes" feature. Also, a native English speaker has been engaged to improve the fluency and the readability of the manuscript.

We believe that the revision proposed by the referees, and further implemented in the text, contributed to improve the manuscript. Thus, we kindly ask you to re-consider the manuscript for publication.

Please find a point-by-point response to the referees’ comments below.

Best regards,

Francesco Di Gennaro

Reviewer 2

This article Human Monkeypox: A Comprehensive and Public Health implications reviewed related study and data of MPX from Pubmed, discussed from its history to the recent outbreak; from its pathophysiology to the diagnosis; from the prevention and treatment to the public health response. The demonstration of the article is comprehensive with humanistic care. There are some questions need to be answered and changed as below.

Response: We thank you very much for the encouraging feedback on our manuscript. We have considered all the your useful suggestions we have implemented the text.

      “Unfortunately, antigen and antibody detection assays do not offer MPX -specific 182 confirmation because orthopoxviruses are serologically cross-reactive. When resources 183 are limited, serology and antigen detection procedures are not recommended for 184 diagnosis or case investigation. Furthermore, recent or past vaccine-based immunization 185 may result in misleading positive findings. In addition, PCR on pharyngeal swabs and 186 seminal fluid may be a good strategy considering the transmission pathways of the virus, 187especially if the patient presents symptoms such as sore throat or penile lesions.”

It is necessary to reference articles to prove antigen and antibody detection assays do not offer MPX -specific confirmation, and recent or past vaccine-based immunization may result in misleading positive findings ( Line185,186)

 Response: Many thanks for your suggestions. We add these three references

Peiró-Mestres A, Fuertes I, Camprubí-Ferrer D, Marcos MÁ, Vilella A, Navarro M, Rodriguez-Elena L, Riera J, Català A, Martínez MJ, Blanco JL; Hospital Clinic de Barcelona Monkeypox Study Group. Frequent detection of monkeypox virus DNA in saliva, semen, and other clinical samples from 12 patients, Barcelona, Spain, May to June 2022. Euro Surveill. 2022 Jul;27(28). doi: 10.2807/1560-7917.ES.2022.27.28.2200503. PMID: 35837964.

Aden TA, Blevins P, York SW, Rager S, Balachandran D, Hutson CL, Lowe D, Mangal CN, Wolford T, Matheny A, Davidson W, Wilkins K, Cook R, Roulo RM, White MK, Berman L, Murray J, Laurance J, Francis D, Green NM, Berumen RA 3rd, Gonzalez A, Evans S, Hudziec M, Noel D, Adjei M, Hovan G, Lee P, Tate L, Gose RB, Voermans R, Crew J, Adam PR, Haydel D, Lukula S, Matluk N, Shah S, Featherston J, Ware D, Pettit D, McCutchen E, Acheampong E, Buttery E, Gorzalski A, Perry M, Fowler R, Lee RB, Nickla R, Huard R, Moore A, Jones K, Johnson R, Swaney E, Jaramillo J, Reinoso Webb C, Guin B, Yost J, Atkinson A, Griffin-Thomas L, Chenette J, Gant J, Sterkel A, Ghuman HK, Lute J, Smole SC, Arora V, Demontigny CK, Bielby M, Geeter E, Newman KAM, Glazier M, Lutkemeier W, Nelson M, Martinez R, Chaitram J, Honein MA, Villanueva JM. Rapid Diagnostic Testing for Response to the Monkeypox Outbreak - Laboratory Response Network, United States, May 17-June 30, 2022. MMWR Morb Mortal Wkly Rep. 2022 Jul 15;71(28):904-907. doi: 10.15585/mmwr.mm7128e1. PMID: 35834423.

An accurate and rapid Real-time PCR approach for human Monkeypox virus diagnosis Tony Wawina-Bokalanga, Nikola Sklenovska, Bert Vanmechelen, Mandy Bloemen, Valentijn Vergote, Lies Laenen, Emmanuel André, Marc Van Ranst, Jean-Jacques Muyembe-Tamfum, Piet Maes medRxiv 2022.06.23.22276033; doi: https://doi.org/10.1101/2022.06.23.22276033

 “Vaccination with first-generation (e.g. Dryvax, Aventis Pasteur Smallpox Vaccine), 225 second-generation (e.g. ACAM2000) and third-generation (IMVAMUNE and LC16m8) 226 vaccines is the first line of defense against orthopoxvirus disease. Despite the fact that 227 these vaccines provide great protection, their widespread use is limited by the high rate 228 of adverse events associated with live, attenuated virus immunization. 229 All first- and second-generation vaccines use live, replication-competent virus. A 230 successful vaccination produces a lesion at the site of administration that generates 231 infectious virus. However, there is risk of adverse events such as autoinoculation to other 232 parts of the body as well as inadvertent transmission to other people, postvaccine 233 encephalitis, and disseminated infection in immunodepressed patients. While third 234 generation vaccine has a higher safety profile owing to the lost ability to replicate in 235 mammalian cells and non-production of a lesion at the site of vaccination, it no longer 236 presents a risk of autoinoculation, inadvertent transmission, or systemic spread. It is 237 reserved for persons with increased risk factors for adverse events such as HIV, 238 haematological and immunosuppressed patients. [34-35]”

I would suggest to discuss about prevention (Line225-238) part before treatment (Line199-224) in “Prevention and Treatment” section

Response: Many thanks we modify following your suggestion and prevention section is before treatment part.

 “In the community, in order to stop the chains of transmission, it would be important 266 to intensify surveillance in specific population groups – such as MSM communities during 267 the current outbreak -, along with case and cluster investigation and contact tracing 268 activities. Indeed, surveillance and case investigation for MPXV in the current context are 269 essential to rapidly identify cases, clusters and sources of infection as early as possible in 270 order to provide optimal clinical care, isolate cases to prevent further transmission, 271 identify and manage contacts, adapt effective prevention methods and control based on 272 the most commonly identified routes of transmission.

Please correct some wording (“MPXV” to “MPX”) and symbol (“current outbreak “- “ to “–“  ) mistakes.

 Response: Many thanks for your suggestions. We modify it and a native English speaker revised the paper.

As for vaccines and immunization, although smallpox infection vaccines are thought 285 to provide protection against MPX, using them during the 2022 outbreak is unlikely. 286 Because of the cessation of smallpox vaccination programs over the last 50 years, as well 287 as a lack of supply of effective vaccines, which are mostly only available in a few national 288 stockpiles, these countermeasures, particularly suitable for contact prophylaxis, are not 289 readily available in the short term.

  I understand it isn’t readily available in the short term. Since smallpox infection vaccines provided protection against MPX, and the technique is ready, it should be encouraged to product those vaccines as soon as possible, at least for specific population groups.

 Response: Many thanks for your comments. We add in conclusion section this important future perspectives as an important goal for scientific community.

The logic of demonstration in each paragraph needs to be improved. There are many paragraphs that saying the same topic could be combined.

Response: We have appreciated the positive feedback to our manuscript

We have considered all your useful suggestions and native English speaker has been engaged to improve the fluency and the readability of the manuscript.  We believe that the revision proposed by the referees, and further implemented in the text, contributed to improve the manuscript.

Reviewer 3 Report

This review on the new outbreak of Monkey pox virus is comprehensive, interesting and well written.

Although other reviews on the same subject have already been published they are different, and focus on other points.

Minor point: the first word line 155 would be "starting" and not "staring" ?

Author Response

To Microorganisms editor,

We have appreciated the positive feedback to our manuscript “Human Monkeypox: A Comprehensive and Public Health implications”.

We have considered all the useful suggestions made by the referees and we have implemented the text. We have also satisfied the technical requirements according to the journal guidelines. Modifications have been highlighted using the "track changes" feature. Also, a native English speaker has been engaged to improve the fluency and the readability of the manuscript.

We believe that the revision proposed by the referees, and further implemented in the text, contributed to improve the manuscript. Thus, we kindly ask you to re-consider the manuscript for publication.

Please find a point-by-point response to the referees’ comments below.

Best regards,

Francesco Di Gennaro

Reviewer 3

This review on the new outbreak of Monkey pox virus is comprehensive, interesting and well written.

Although other reviews on the same subject have already been published they are different, and focus on other points.

Minor point: the first word line 155 would be "starting" and not "staring" ?

Response: We thank you very much for the encouraging feedback on our manuscript. A native English speaker has been engaged to improve the fluency and the readability of the manuscript.

Round 2

Reviewer 1 Report

The manuscript is ready for publication.